# Within-host diversity improves phylogenetic and transmission reconstruction of SARS-CoV-2 outbreaks

Arturo Torres Ortiz[1,2]*, Michelle Kendall[3], Nathaniel Storey[4], James Hatcher[4], Helen Dunn[4], Sunando Roy[2], Rachel Williams[5], Charlotte Williams[5], Richard A Goldstein[5], Xavier Didelot[3], Kathryn Harris[4,6], Judith Breuer[2], Louis Grandjean[2]*

[1]Department of Infectious Diseases, Imperial College London, London, United Kingdom; [2]Department of Infection, Immunity and Inflammation, University College London, London, United Kingdom; [3]Department of Statistics, University of Warwick, Coventry, United Kingdom; [4]Department of Microbiology, Great Ormond Street Hospital, London, United Kingdom; [5]UCL Genomics, University College London, London, United Kingdom; [6]Department of Virology, East & South East London Pathology Partnership, Royal London Hospital, Barts Health NHS Trust, London, United Kingdom

*For correspondence:
a.ortiz@ucl.ac.uk (ATO);
l.grandjean@ucl.ac.uk (LG)

**Competing interest:** The authors declare that no competing interests exist.

**Abstract** Accurate inference of who infected whom in an infectious disease outbreak is critical for the delivery of effective infection prevention and control. The increased resolution of pathogen whole-genome sequencing has significantly improved our ability to infer transmission events. Despite this, transmission inference often remains limited by the lack of genomic variation between the source case and infected contacts. Although within-host genetic diversity is common among a wide variety of pathogens, conventional whole-genome sequencing phylogenetic approaches exclusively use consensus sequences, which consider only the most prevalent nucleotide at each position and therefore fail to capture low-frequency variation within samples. We hypothesized that including within-sample variation in a phylogenetic model would help to identify who infected whom in instances in which this was previously impossible. Using whole-genome sequences from SARS-CoV-2 multi-institutional outbreaks as an example, we show how within-sample diversity is partially maintained among repeated serial samples from the same host, it can transmitted between those cases with known epidemiological links, and how this improves phylogenetic inference and our understanding of who infected whom. Our technique is applicable to other infectious diseases and has immediate clinical utility in infection prevention and control.

## Editor's evaluation

This valuable study presents a novel and theoretically interesting model to account for viral diversity within hosts in evolutionary and genomic analyses of pathogens. The simulation results are solid and suggest that the model may provide new insight into SARS-CoV-2's and other pathogens' evolutionary dynamics.

## Introduction

Understanding who infects whom in an infectious disease outbreak is a key component of infection prevention and control (*Didelot et al., 2012*). The use of whole-genome sequencing allows for

**eLife digest** During an infectious disease outbreak, tracing who infected whom allows public health scientists to see how a pathogen is spreading and to establish effective control measures. Traditionally, this involves identifying the individuals an infected person comes into contact with and monitoring whether they also become unwell. However, this information is not always available and can be inaccurate.

One alternative is to track the genetic data of pathogens as they spread. Over time, pathogens accumulate mutations in their genes that can be used to distinguish them from one another. Genetically similar pathogens are more likely to have spread during the same outbreak, while genetically dissimilar pathogens may have come from different outbreaks. However, there are limitations to this approach. For example, some pathogens accumulate genetic mutations very slowly and may not change enough during an outbreak to be distinguishable from one another. Additionally, some pathogens can spread rapidly, leaving less time for mutations to occur between transmission events.

To overcome these challenges, Torres Ortiz et al. developed a more sensitive approach to pathogen genetic testing that took advantage of the multiple pathogen populations that often coexist in an infected patient. Rather than tracking only the most dominant genetic version of the pathogen, this method also looked at the less dominant ones.

Torres Ortiz et al. performed genome sequencing of SARS-CoV-2 (the virus that causes COVID-19) samples from 451 healthcare workers, patients, and patient contacts at participating London hospitals. Analysis showed that it was possible to detect multiple genetic populations of the virus within individual patients. These subpopulations were often more similar in patients that had been in contact with one another than in those that had not. Tracking the genetic data of all viral populations enabled Torres Ortiz et al. to trace transmission more accurately than if only the dominant population was used.

More accurate genetic tracing could help public health scientists better track pathogen transmission and control outbreaks. This may be especially beneficial in hospital settings where outbreaks can be smaller, and it is important to understand if transmission is occurring within the hospital or if the pathogen is imported from the community. Further research will help scientists understand how pathogen population genetics evolve during outbreaks and may improve the detection of subpopulations present at very low frequencies.

detailed investigation of disease outbreaks, but the limited genetic diversity of many pathogens often hinders our understanding of transmission events (*Campbell et al., 2018*). As a consequence of the limited diversity, many index case and contact pairs will share identical genotypes, making it difficult to ascertain who infected whom.

Within-sample genetic diversity is common among a wide variety of pathogens (*Mongkolrattan-othai et al., 2011*; *Lieberman et al., 2016*; *Dinis et al., 2016*; *Leitner, 2019*; *Popa et al., 2020*). This diversity may be generated de novo during infection, by a single transmission event of a diverse inoculum or by independent transmission events from multiple sources (*Worby et al., 2014*). The maintenance and dynamic of within-host diversity is then a product of natural selection, genetic drift, and fluctuating population size (*Didelot et al., 2012*). The transmission of within-host variation between individuals is also favored as a large inoculum exposure is more likely to give rise to infection (*Murphy et al., 1984*; *Han et al., 2019*; *Lee et al., 2022*; *Sender et al., 2021*; *Spinelli et al., 2021*; *Trunfio et al., 2021*). The amount of within-sample diversity transmitted from index case to contact is determined by the bottleneck size (*Zwart and Elena, 2015*), with stringent bottlenecks limiting the number of genotypes transmitted from the host to the recipient, and wide bottlenecks allowing for the transmission of higher levels of genetic diversity (*Worby et al., 2014*).

Phylogenetic analysis provides information regarding the structure of the genetic diversity among pathogen isolates. Moreover, pathogen phylogenetic trees can be used as input for many downstream analysis, including inference of transmission events, population size dynamics, or estimation of parameters of epidemiological models (*Didelot et al., 2018*). Most genomic and phylogenetic workflows involve either genome assembly or alignment of sequencing reads to a reference genome. In both cases, conventionally the resulting alignment exclusively represents the most common

nucleotide at each position. This is often referred to as the consensus sequence. Although genome assemblers may output contigs (combined overlapping reads) representing low-frequency haplotypes, only the majority contig is kept in the final sequence. In a mapping approach, a frequency threshold for the major variant is usually pre-determined, under which a position is considered ambiguous. The lack of genetic variation between temporally proximate samples and the slow mutation rate of many pathogens results in direct transmission events sharing exact sequences between the hosts when using the consensus sequence approach. For instance, the substitution rate of SARS-CoV-2 has been inferred to be around two mutations per genome per month (*Harvey et al., 2021*). Given its infectious period of 6 days (*Byrne et al., 2020*), most consensus sequences in a small-scale outbreak will show no variation between them. This lack of resolution and poor phylogenetic signal complicate phylogenetic inference, limiting the downstream analysis and conclusions that can be extracted from the phylogenetic tree. Previous work has shown the advantages of using within-host diversity to infer transmission events compared to using consensus sequences (*Wymant et al., 2018*; *De Maio et al., 2018*). Aside from transmission inference, the use of the within-host pathogen genetic data directly within phylogenetic inference will improve any downstream analysis using a phylogenetic tree as starting point.

We hypothesize that the failure of consensus sequence approaches to capture within-sample variation arbitrarily excludes meaningful data and limits pathogen phylogenetic and transmission inference, and that including within-sample diversity in a phylogenetic model would significantly increase the evolutionary and temporal signal and thereby improve our ability to infer infectious disease phylogenies and transmission events.

We tested our hypothesis on multi-institutional SARS-CoV-2 outbreaks across London hospitals that were part of the COVID-19 Genomics UK (COG-UK) consortia (*COVID-19 Genomics UK COG-UK, 2020*). Technical replicates, repeated longitudinal sampling from the same patient, and epidemiological data allowed us to evaluate the presence and stability of within-sample diversity within the host and in independently determined transmission chains. We also evaluated the use of within-sample diversity in phylogenetic analysis by conducting outbreak and phylogenetic simulations of sequencing data using a phylogenetic model that accounts for the presence and transmission of within-sample variation. We show the effects on phylogenetic inference of using consensus sequences in the presence of within-sample diversity, and propose that existing phylogenetic models can leverage the additional diversity given by the within-sample variation and reconstruct the phylogenetic relationship between isolates. Lastly, we show that by taking into account within-sample diversity in a phylogenetic model, we improve the temporal signal in SARS-CoV-2 outbreak analysis. Using both phylogenetic outbreak reconstruction and simulation, we show that our approach is superior to the current gold standard whole-genome consensus sequence methods.

## Results

### Sampling, demographics, and metadata

Between March 2020 and November 2020, 451 healthcare workers, patients, and patient contacts at the participating North London Hospitals were diagnosed at the Camelia Botnar Laboratories with SARS-CoV-2 by PCR as part of a routine staff diagnostic service at Great Ormond Street Hospital NHS Foundation Trust (GOSH). A total of 289 isolates were whole-genome sequenced using the Illumina NextSeq platform, which resulted in 522 whole-genome sequences including longitudinal and technical replicates (*Supplementary file 1*). The mean participant age was 40 years of age (median 38.5 years of age, interquartile range [IQR] 30–50 years of age), and 60% of the participants were female (*Supplementary file 2*). All samples were SARS-CoV-2 positive with real-time qPCR cycle threshold ($C_t$) values ranging from 16 to 35 cycles (*Supplementary file 2*). The earliest sample was collected on March 26, 2020, while the latest one dated to November 4, 2020 (*Figure 1—figure supplement 1a*). A total of 291 samples had self-reported symptom onset data, for which the mean time from symptom onset to sample collection date was 5 days (IQR 2–7 days, *Figure 1—figure supplement 1b*). More than 90% of the samples were taken from hospital staff, while the rest comprised patients and contacts of either the patients or the staff members (*Supplementary file 2*).

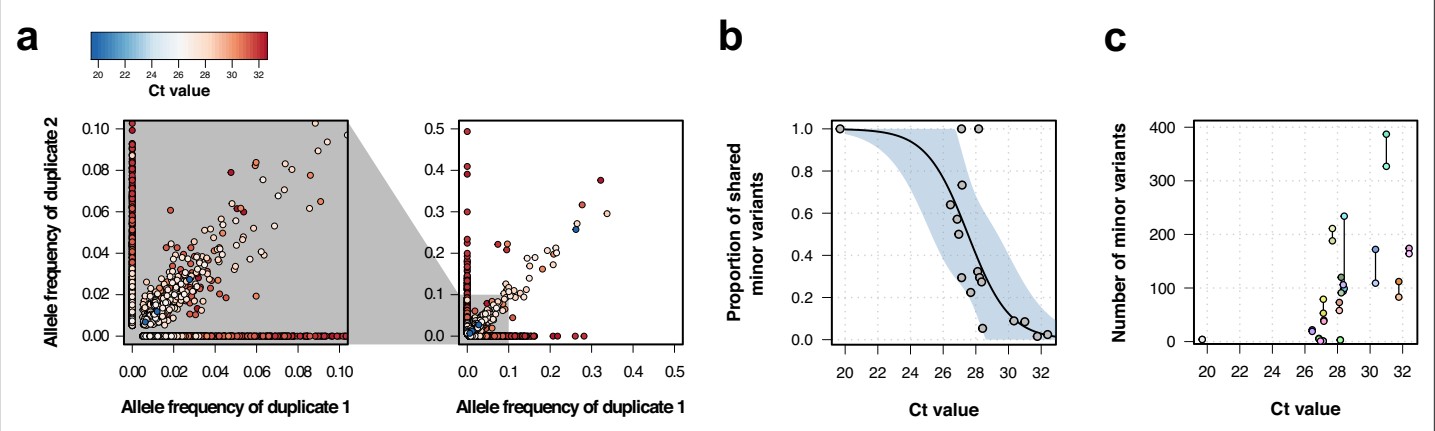

**Figure 1.** Genomic analysis of technical duplicates before filtering. (**a**) Allele frequency comparison between technical replicates for all frequencies (right) and for frequencies up to 1% (left). Colors represent the $C_t$ value for the sample. (**b**) Proportion of shared minor variants between technical replicates in relation to the $C_t$ value. (**c**) Total number of minor variants in relation to the $C_t$ value. Lines linked two technical replicates. Each sequence has a different color, with sequences from the same patient having a different shade of the same color.

The online version of this article includes the following figure supplement(s) for figure 1:

**Figure supplement 1.** Collection date distribution and time from symptom and days from symptom onset.

**Figure supplement 2.** Sample mean coverage distribution.

**Figure supplement 3.** Effects of $C_t$ value on whole-genome sequencing data.

**Figure supplement 4.** Proportion of shared minor variants between technical replicates using different filters of allele frequency.

## Genomic analysis of SARS-CoV-2 sequences

A total of 454 whole genomes with mean coverage higher than 10× were kept for further analysis, resulting in an average coverage across isolates of 2457× (*Figure 1—figure supplement 2*). Allele frequencies were extracted using the pileup functionality within *bcftools* (*Danecek et al., 2011*) with a minimum base and mapping quality of 30, which represents a base call error rate of 0.1%. Variants at low frequency at positions where the mapped reads support more than one allele were coined as minor or low-frequency variants. Variants were filtered further for read position bias and strand bias. Only minor variants with an allele frequency of at least 2% were kept as putative variants. Samples with a frequency of missing bases higher than 10% were excluded, keeping 350 isolates for analysis. The mean number of low-frequency variants was 12 (median 3, IQR 1.00–9.75), although both the number of variants and its deviation increased at high $C_t$ values (*Figure 1—figure supplement 3*).

## Within-sample variation in technical replicates

To understand the stability of within-sample variation and minimize spurious variant calls, we sequenced and analyzed technical replicates of 17 samples. Overall, when the variant was present in both duplicates the correlation of the variant frequencies was high ($R^2$=0.9, *Figure 1a*, right). The high correlation was also maintained at low variant frequencies (*Figure 1a*, left).

Minor variants were less likely to be shared when one or more of the paired samples had low viral load. These discrepancies may appear because of amplification bias caused by low genetic material, base calling errors due to low coverage, or low base quality. The mean proportion of discrepant within-sample variants between duplicated samples was 0.39 (sd = 0.29), although this varied between duplicates (*Figure 1—figure supplement 4*). $C_t$ values in RT-PCR obtained during viral amplification are inversely correlated with viral load (*Tom and Mina, 2020*). The proportion of shared intra-host variants was negatively correlated with $C_t$ values in a logistic model (estimate = −0.78, p-value = 0.008), with higher $C_t$ values associated with a lower amount of shared intra-host variants (*Figure 1c*). The number of within-sample variants detected also increased with $C_t$ value, as well as the deviation in the number of variants between duplicates (*Figure 1d*). This could be explained either by an increase in the number of spurious variants at low viral loads (*Tonkin-Hill et al., 2021*), biased amplification of low-level subpopulations minor rare alleles (*McCrone and Lauring, 2016*), or due to the accumulation

**Table 1.** SNP distance between pairs of samples.

| Sample relationship | Estimate (95% CI) | p-Value |
|---|---|---|
| None | 11.04 (10.94–11.15) | Reference |
| Hospital | 9.78 (9.48–10.09) | $<1\times10^{-4}$ |
| Department | 5.15 (4.54–5.83) | $<1\times10^{-4}$ |
| Epidemiological | 1.5 (1.22–1.78) | $<1\times10^{-4}$ |
| Longitudinal duplicates | 0 (0–0.2) | $<1\times10^{-4}$ |
| Technical replicate | 0 (0–0.2) | $<1\times10^{-4}$ |

of within-host variation through time, as late stages of infection are usually characterized by high $C_t$ values (low viral load).

Based on these results, only samples with a $C_t$ value equal or lower than 30 cycles were considered, which resulted in 249 samples kept for analysis. Additionally, only variants with a frequency higher or equal than 2% were used. For the filtered dataset, 414 out of 29,903 positions were polymorphic for the consensus sequence, while the alignment with within-sample diversity had 1039 SNPs. Of these, 699 positions had intra-host diversity, of which 78% (549/699) were singletons. The majority of samples (207/249, 83%) contained at least one position with a high-quality within-host variant, and the median amount of intra-host variants per sample was 2 (IQR 1–4.5).

## Within-sample variation in epidemiologically linked samples

Given the limited genomic information in the consensus sequences, epidemiological data is often necessary to infer the directionality of transmission. We categorized our samples within the following groups: samples that (a) did not have any recorded epidemiological link, (b) were from the same hospital (possibly linked), (c) were part of the same department within the same hospital (probable link), (d) had an epidemiological link within the same department of the same hospital (proven link), (e) were a longitudinal replicate from the same patient, and (f) a technical replicate from the same sample.

We tested the concordance between epidemiological and genomic data by determining the SNP distance between pairs of samples with epidemiological links and without them. Pairs of samples from the same hospital, department, epidemiologically linked, or longitudinal and technical replicates had a lower SNP distance (were more closely related) than those samples that did not have any relationship, although this difference was small in the case of pairs of samples from the same hospital (*Table 1*).

To understand the distribution of shared low-frequency variants among different groups of samples, we performed a pairwise comparison of all samples and calculated the proportion of shared within-sample variants (shared variants divided by total variants in the pair) within groups with epidemiological links and without them. The proportion of shared within-host variants was higher between technical replicates, longitudinal duplicates, epidemiologically linked samples, and samples taken from individuals from the same department when compared to pairs with no epidemiological links, although the range of this probability was large (*Figure 2*, *Figure 2—figure supplement 1*). The probability of sharing a low-frequency variant was inferred using a logistic regression model (*Figure 2—figure supplement 2*). There was a tendency for the probability to increase with variant frequency, but the association was not strong (odds ratio 1.8, 95% CI 0.9–3.5, p=0.08). The probability of sharing a low-frequency variant for samples with no epidemiological links was $9.5 \times 10^{-6}$ (95% CI $8.8 \times 10^{-6}$ – $1.02 \times 10^{-5}$). Samples from the same hospital did not have a probability significantly higher than those without any link ($3.3 \times 10^{-3}$, 95% CI $2.7 \times 10^{-3}$ – $4.03 \times 10^{-3}$). On the other hand, pairs from the same department, with epidemiological links, replicates, or technical replicates all had a significantly higher probability of sharing a low-frequency variant when compared to those pairs with no link (all Wald test p-values <0.001). The inferred probabilities for pairs from the sample department was 1.4% (95% CI 0.9–2.1%), which increased to 5% for pairs with epidemiological links (95% CI 4.2–6.4%). For longitudinal replicates, the probability was inferred to be 38% (95% CI 35–41%), and were shared between multiple time points (*Figure 2—figure supplement 3*). Technical replicates were estimated to have the highest probability (70%, 95% CI 64–76%).

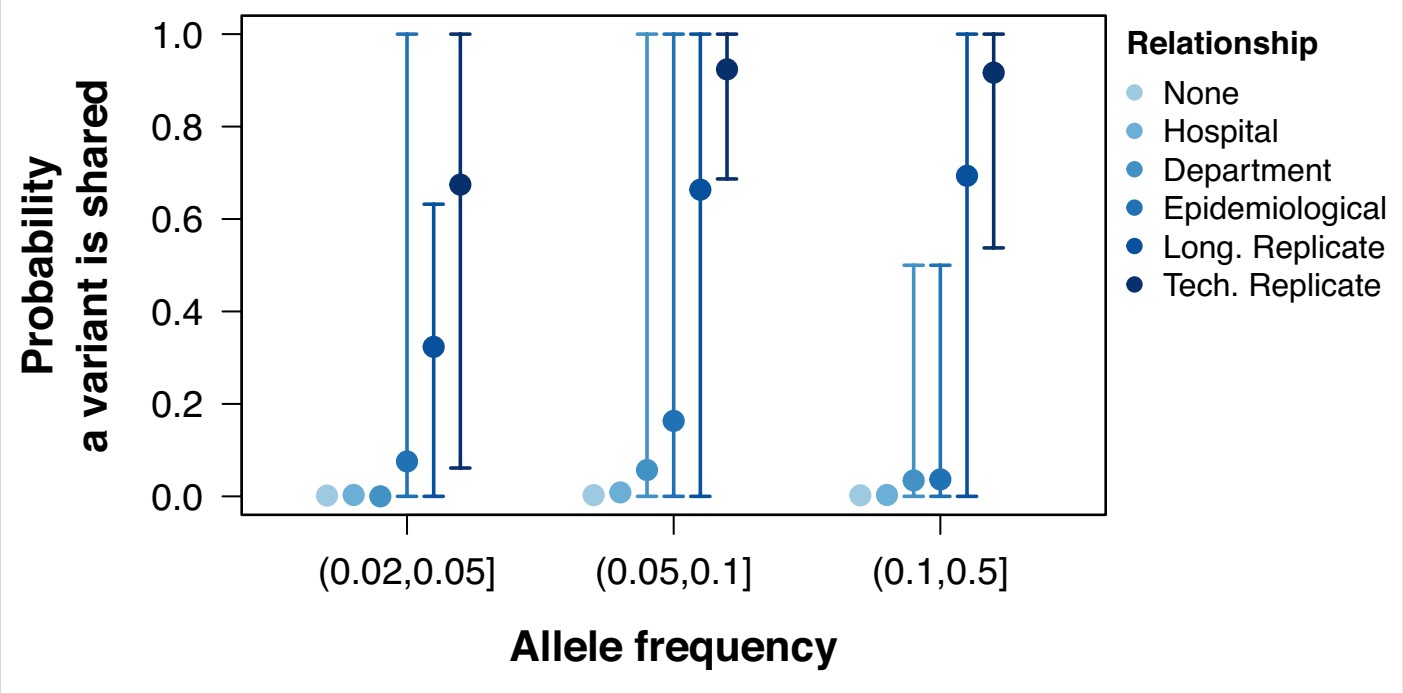

**Figure 2.** Probability of sharing within-host variants in sample pairs. The probability of variants shared between pairs of samples calculated as the number of low-frequency variants in both samples divided by the total number of variants between the pair. Colors grouped samples by their relationship. Points represent the mean probability a variant is shared between all pairwise samples within a group and allele frequency. Error bars show the 95th and 5th percentiles.

The online version of this article includes the following figure supplement(s) for figure 2:

**Figure supplement 1.** Allele frequency comparison in pairwise sample pairs.

**Figure supplement 2.** Probability that minor variants are shared.

**Figure supplement 3.** Dynamics of low-frequency variants in longitudinal duplicates.

## Within-host diversity model outperforms the consensus model in simulations

The effect of within-sample diversity in phylogenetic inference was tested by evaluating the accuracy in the reconstruction of known phylogenetic trees using a conventional phylogenetic model and a model that accounts for within-sample variation.

The presence of within-sample diversity was coded in the genome alignment using existing IUPAC nomenclature (*IUPAC-IUB Joint Commission on Biochemical Nomenclature (JCBN), 1984*). For the consensus sequence alignment, only the four canonical nucleotides were used (*Figure 3a and b*), while the proposed alignment retained the major and minor allele information as independent character states (*Figure 3c and d*).

In order to evaluate the differences in tree inference with and without the inclusion of within-sample diversity, we simulated genome alignments for 100 random trees using a phylogenetic model where both major and minor variant combinations were considered, resulting in a total of 16 possible states (*Figure 3d*). In the proposed model, transitions and transversions between the four nucleotides in the population occur in the following steps: first a minority variant evolves at low frequency, then the minor variant increases its frequency to become the majority nucleotide, and finally the variant is fixed (*Figure 3d*), with all the steps being reversible. Therefore, within-host evolutionary dynamics are modeled by explicitly considering base change as a process of minor variant evolution and eventual fixation. The substitution rates chosen for the simulations, as shown in *Supplementary file 4*, were selected to reflect a slow rate of minor variant evolution and a fast rate at which minor variants are lost or fixated in the population, which in turn results in a highly dynamic landscape of within-sample variation, with the four canonical nucleotides 100 times more likely to be present than low-frequency variants.

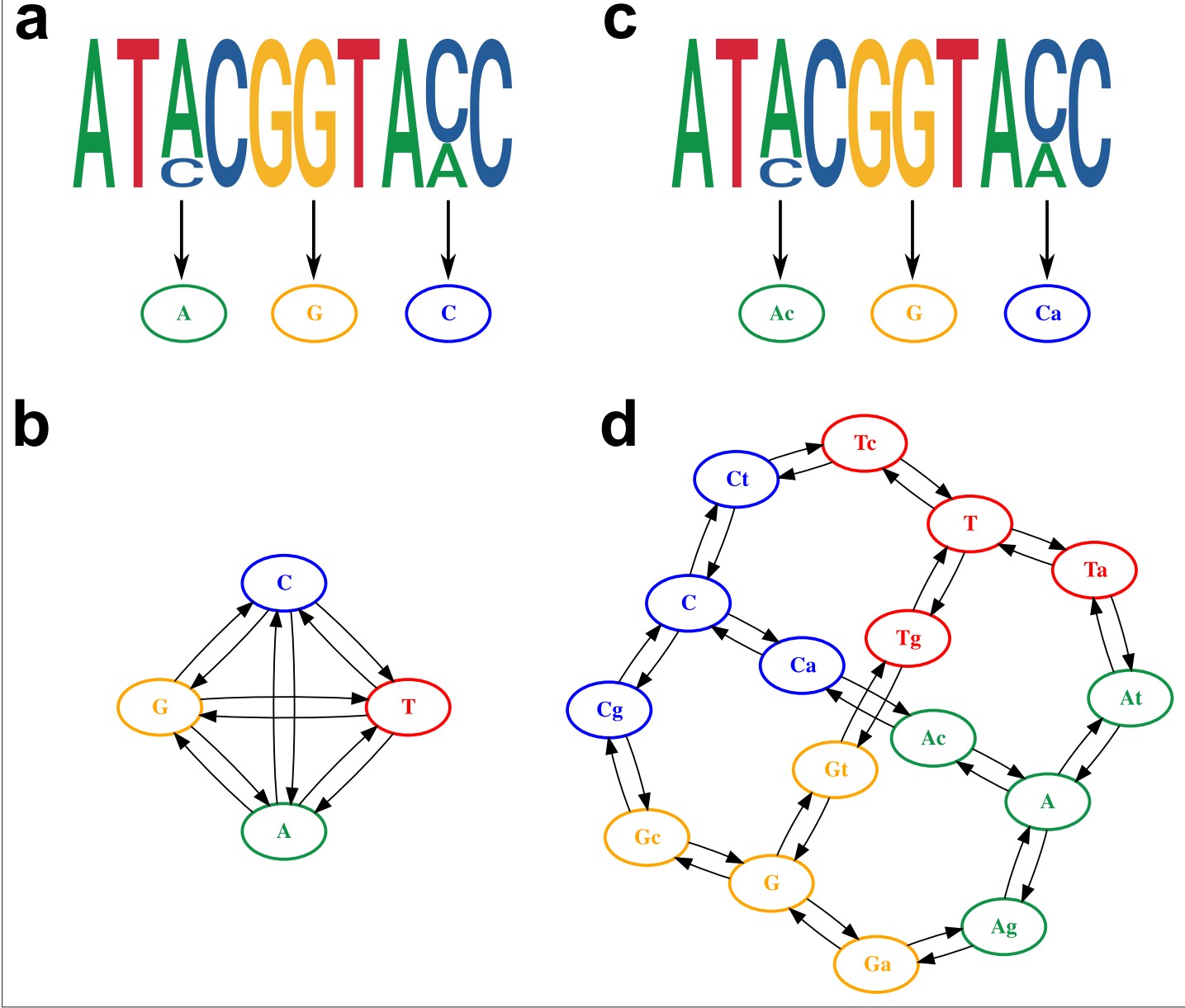

**Figure 3.** Model of within-host diversity. Proposed evolutionary model of within-host diversity in genomic sequences. Uppercase letters represent the major variant in the population, while lowercase letters indicate presence of a minor variant alongside the major one. (**a, c**) Genome sequences where some positions show within-sample variation (top), represented by a major allele (big size letter) and a minor one (smaller size), as well as its representation in the alignment (bottom). (**b, d**) Models of nucleotide evolution. Character transitions are indicated by arrows. (**a**) Consensus sequence, where only the major allele is represented in the alignment. (**b**) Model of nucleotide evolution using the consensus sequence, with four character states representing the four nucleotides. (**c**) Sequence with within-sample variation, represented by an uppercase letter for the major allele and a lower case letter for the minor allele. (**d**) Model of nucleotide evolution with 16-character states accounting for within-sample variation.

From the simulated genomes, two types of alignments were generated: a consensus sequence, where only the major allele was considered (*Figure 3a*); and an alignment that retained the major and minor allele information as independent character states (*Figure 3c*). From the simulated alignments, RaxML-NG was used to infer phylogenetic trees (*Kozlov et al., 2019*). The consensus sequence was analyzed with a GTR+$\gamma$ model, while the PROTGTR+$\gamma$ model was used in order to accommodate the extra characters of the alignment with within-sample diversity and major/minor variant information.

The two models were evaluated for their ability to infer the known phylogeny that included within-host diversity. The estimated phylogenies were compared to the known tree using different measures

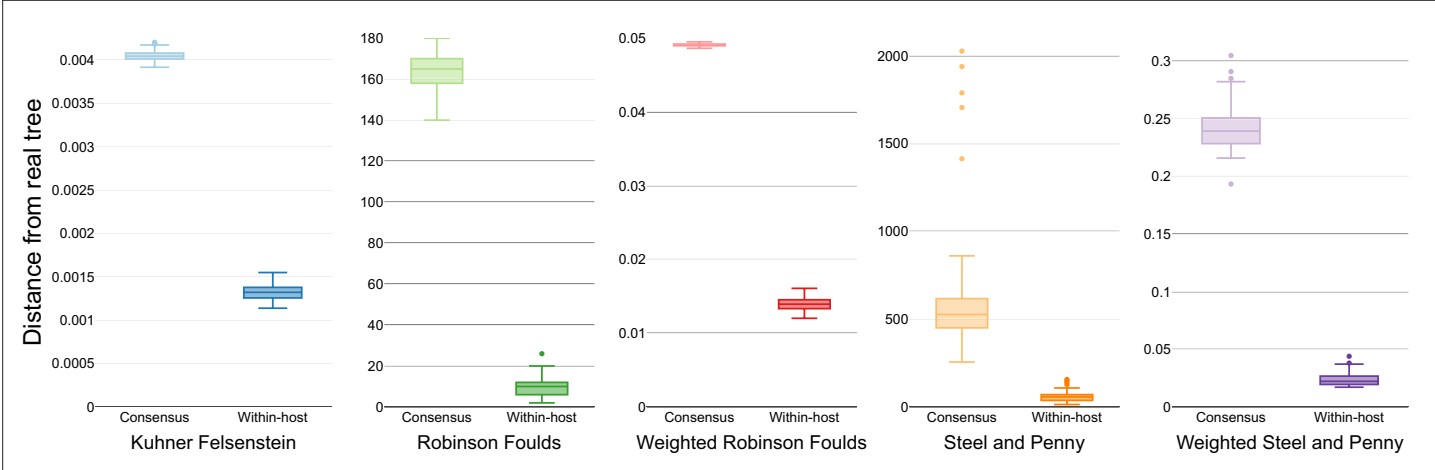

**Figure 4.** Similarity scores for inferred trees. Comparison of the phylogenetic trees inferred using simulated sequences from known random starting trees and different phylogenetic models. Colors differentiate the metrics used for the comparison.

The online version of this article includes the following figure supplement(s) for figure 4:

**Figure supplement 1.** Similarity scores for inferred trees with different rates.

**Figure supplement 2.** Similarity scores for inferred trees from coalescent simulations.

to capture dissimilarities in a variety of aspects relevant to tree inference (*Supplementary file 3*). For all the metrics employed, the phylogenies inferred explicitly using within-host diversity as independent characters approximated better to the initial tree than the one using the consensus sequence (*Figure 4*). Additionally, the transition/transversion rates inferred by the phylogenetic models accounting for within-host diversity accurately reflect the rates used for the simulation of genomic sequences (*Supplementary files 4–6*).

As different pathogens are likely to show different dynamics of within-host variation and the rates used for the simulations will inevitably affect the improvement of using the 16-state model, we simulated genomes with different parameters. As expected, choosing rates that promote an abundant and stable landscape of low-frequency variation (rate of minor variant acquisition of 20, and rates of variant switch and lost of 1) made the 16-state model to perform better than the model using consensus sequences, which improved as the proportion of low-frequency variants decreased (*Figure 4—figure supplement 1*). Conversely, in simulations using a Jukes-Cantor DNA model, and therefore without any low-frequency variation, both models showed similar results (*Figure 4—figure supplement 1*).

To understand the effects of genetic linkage between sites in the phylogenetic model due to the clonal relationships between genomes, we evaluated another set of simulations where the starting tree was generated using the coalescent model, which increases the correlation between sites. For all metrics used, the model using low-frequency variants inferred phylogenies more similar to the starting coalescent tree than those inferred using the consensus sequence (*Figure 4—figure supplement 2*).

We further assessed the effect of within-host diversity in phylogenetic inference by simulating pathogen evolution throughout the time frame of infectious disease outbreaks (*De Maio et al., 2018*). We simulated outbreaks using TransPhylo (*Didelot et al., 2017*) with a host population varying between 10 and 15 hosts, no recombination, complete sampling of the outbreak, and selecting epidemiological parameters to match the transmission dynamics of SARS-CoV-2. For each outbreak, we simulated the evolution and transmission of the pathogen population within each host with varying values of mutation rates and transmission bottlenecks using fastsimcoal2 (*Excoffier et al., 2013*) as previously described by *De Maio et al., 2018*. We compared the resulting phylogenetic trees to the real outbreak phylogeny using the Kuhner-Felsenstein distance (*Kuhner and Felsenstein, 1994*). Even though using consensus sequences performed better than a random distribution of trees, using within-host diversity outperformed the consensus sequence in all instances (*Figure 5*). The phylogenies inferred using within-host diversity were more similar to the real outbreak phylogeny for wider bottleneck sizes, with the best performance when no bottleneck was present. As expected, both the consensus sequences and the sequences reflecting within-sample diversity were more informative at

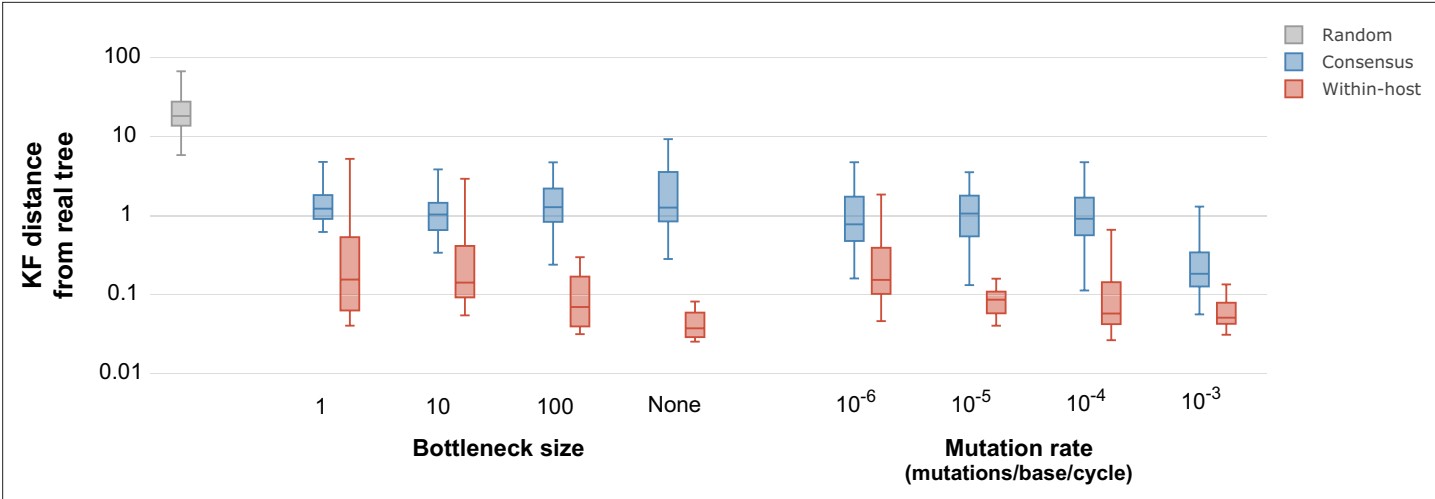

**Figure 5.** Inferred phylogenetic trees from outbreak simulations. Kuhner and Felsenstein (KF) tree distance between phylogenetic trees from simulated outbreaks. Phylogenies were inferred using consensus sequences (blue) and alignments reflecting within-sample diversity (red), and compared to the known phylogeny of the simulated outbreak. For reference, gray color represents a set of random trees. Outbreak simulations were performed with different bottleneck sizes and mutation rates. The mutation rate is measured as the number of mutations per base per generation cycle.

higher mutation rates, even though the consensus sequence only showed improvement with a mutation rate of $10^{-3}$ mutations per base per generation cycle (***Figure 5***).

## Within-host diversity improves the resolution in SARS-CoV-2 phylogenetics

Genome sequences collected at different time points are expected to diverge as time progresses, resulting in a positive correlation between the isolation date and the number of accumulated mutations (temporal signal) (***Rieux and Balloux, 2016***). The alignment with consensus sequences and the one reflecting within-sample variation were used to infer two different phylogenetic trees (***Figure 6—figure supplement 1***). Longitudinal samples in the phylogeny inferred using within-host diversity reflected the expected temporal signal, with an increase in genetic distance as time progressed between the longitudinal pairs in a linear model (coefficient 2.24, 0.59–3.88 95% CI, p=0.019, ***Figure 6—figure supplement 2***). The difference in $C_t$ value among longitudinal duplicates was not correlated with higher genetic distances (coefficient 1.62, –0.66 to 3.91 95% CI, p=0.2). Similarly, we analyzed the number of low-frequency variants within outbreaks by counting the number of within-sample variants for each isolate belonging to a specific outbreak and inferred their change through time taking the earliest isolate date as the starting point of the outbreak. In general, as the outbreaks progressed the number of low-frequency variants increased (coefficient 0.16, 0.06–0.27 95% CI, p=0.003, $r^2$=0.19, ***Figure 6—figure supplement 3***).

We analyzed the impact of using within-sample variation on the temporal structure of the phylogeny by systematically identifying clusters of tips in the phylogenetic tree with an identical consensus sequence and no temporal signal. We then performed a root-to-tip analysis using the tree inferred with intra-sample diversity. Only clusters with more than three tips were used for the root-to-tip analysis. The majority of clusters (10/11) showed a positive correlation between the distance of the tips to the root and the collection dates, demonstrating a significant temporal signal between samples when there was none using the conventional consensus tree (***Figure 6***).

To illustrate the downstream application of the improved phylogenetic resolution, we inferred a time-calibrated phylogeny from the phylogeny inferred using the 16-character state model with the collection dates of the tips using BactDating (***Didelot et al., 2018***; ***Figure 7—figure supplement 1***) and calculated the likelihood of transmission events within potential epidemiologically identified outbreaks using a susceptible-exposed-infectious-removed (SEIR) model (***Lekone and Finkenstädt, 2006***; ***Eldholm et al., 2016***). The SEIR model was parameterized with an average latency period of 5.5 days (***Xin et al., 2022***), an infectious period of 6 days (***Byrne et al., 2020***), and a within-host coalescent rate of 5 days as previously estimated for SARS-CoV-2 (***Wang et al., 2020***). The likelihood

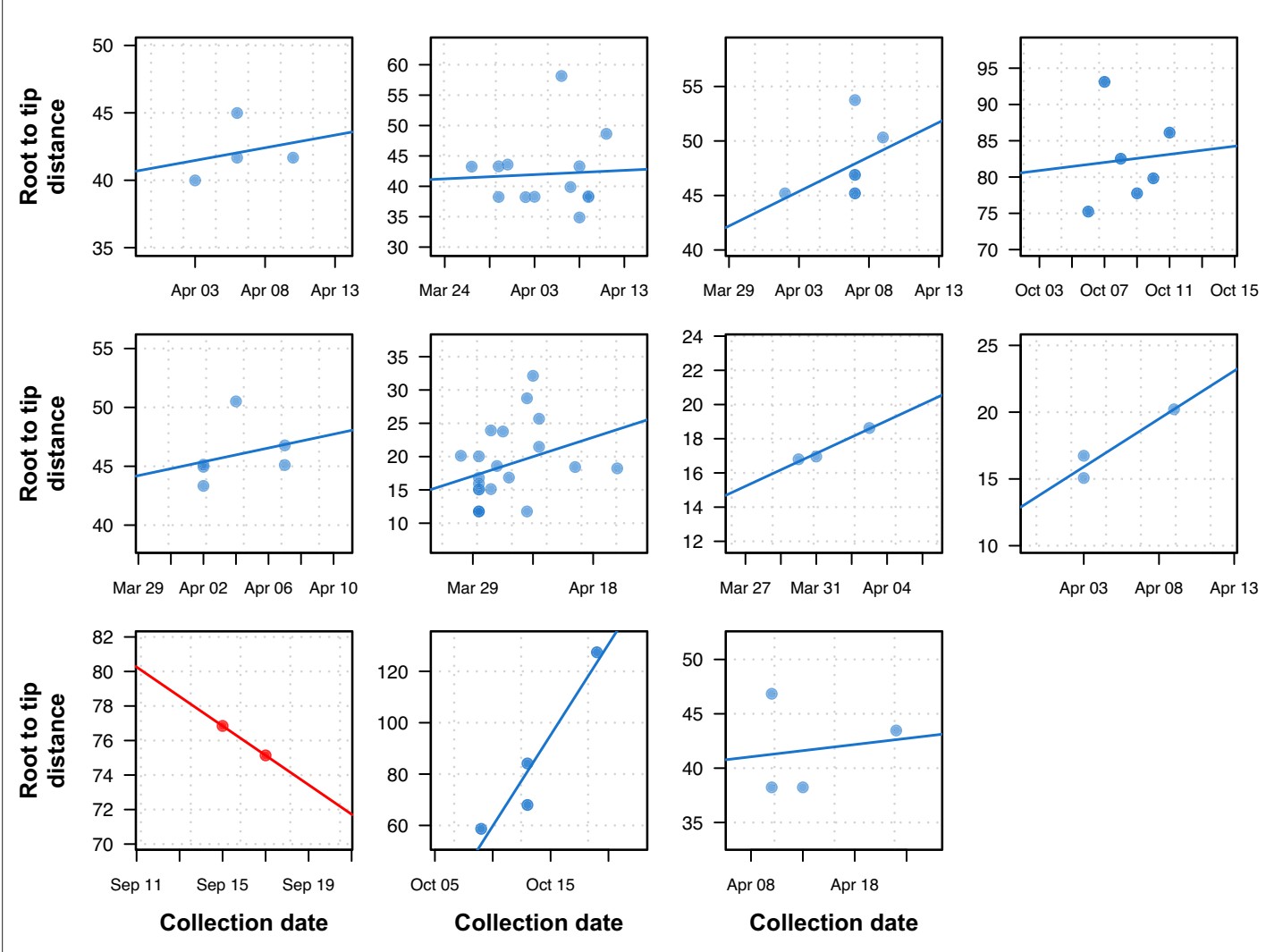

**Figure 6.** Previously uninformative clusters present temporal signal when using within-sample diversity. A set of 11 outbreak clusters (one per panel, each plotting the root-to-tip distance in number of substitutions per genome against time) in which all samples had identical consensus genomes sequences (and therefore no temporal signal). Blue colors indicate those regressions that after utilizing within sample diversity now have a positive slope (temporal signal), and red shows those regressions that have a negative slope (misleading or false positive temporal signal).

The online version of this article includes the following figure supplement(s) for figure 6:

**Figure supplement 1.** Phylogenetic trees for SARS-CoV-2.

**Figure supplement 2.** Genetic distance between longitudinal samples.

**Figure supplement 3.** Number of low-frequency variants within outbreaks as the outbreak progresses.

of transmission was calculated for every pair of samples, while the Edmonds algorithm as implemented in the R package *RBGL* (*Carey et al., 2021*) was used to infer the graph with the optimum branching (*Figure 7c and d*; *Figure 7—figure supplement 2*).

*Figure 7* represents an example of an outbreak involving four hosts, with one patient, one patient contact, and two hospital staff members. All samples have one technical replicate, while patient sample also has two serial samples (which were removed for transmission inference). The ML tree inferred using the consensus sequences (*Figure 7a*, left) shows that most isolates have the exact same consensus sequence. Although this suggests that all isolates belong to the same outbreak, the similarity between sequences precludes exact transmission inference. However, the ML tree inferred using sequences with low-frequency variants correctly clusters technical and longitudinal replicates, and groups the isolates in distinctive sets that better inform transmission inference (*Figure 7b and c*). We

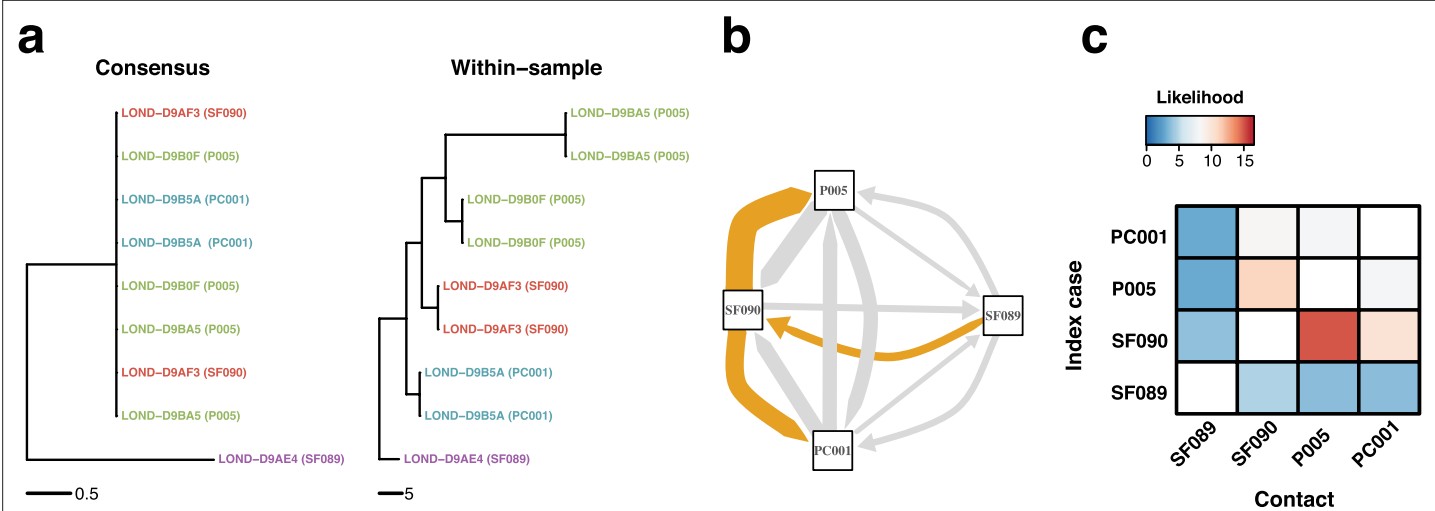

**Figure 7.** Within-sample variation improves resolution of infectious disease outbreaks. Effect of using low-frequency variants in phylogenetic inference. (**a**) Maximum likelihood phylogeny using the consensus sequences (left) and the alignment leveraging within-sample variation. Replicates of the same sample share the same color. Sample IDs are coded as follows: SF, for staff members; P, for patients; and PC, for patient contacts. (**b**) Transmission network inferred using within-sample variation. Edge width is proportional to the likelihood of direct transmission using a susceptible-exposed-infectious-removed (SEIR) model. Colored edges represent the Edmunds optimum branching and thus the most likely chain. (**c**) Heatmap of the likelihood of direct transmission between all pairwise pairs of samples using a SEIR model. Vertical axis is the infector while the horizontal axis shows the infectee.

The online version of this article includes the following figure supplement(s) for figure 7:

**Figure supplement 1.** Time-calibrated phylogenetic trees for SARS-CoV-2.

**Figure supplement 2.** Phylogenetic and transmission for SARS-CoV-2 outbreaks.

applied the same analysis to other potential outbreaks and obtained similar results (*Figure 7—figure supplement 2*).

## Discussion

Detailed investigation of transmission events in an infectious disease outbreak is a prerequisite for effective prevention and control. Although whole-genome sequencing has transformed the field of pathogen genomics, insufficient pathogen genetic diversity between cases in an outbreak limits the ability to infer who infected whom. Using multi-hospital SARS-CoV-2 outbreaks and phylogenetic simulations, we show that including the genetic diversity of subpopulations within a clinical sample improves phylogenetic reconstruction of SARS-CoV-2 outbreaks and determines the direction of transmission when using a consensus sequence approach fails to do so.

The majority of samples sequenced harbored variants at low frequency. However, most variants were not consistently called in technical replicates, suggesting they were spurious or unreliable. Within-sample variation was less consistent between paired technical replicates with lower viral load (higher $C_t$). This is likely to be a consequence of low starting genetic material giving rise to amplification bias during library preparation and sequencing. Establishing a cut-off for high $C_t$ values is therefore important to accurately characterize within-host variation. In our study, we excluded samples with a $C_t$ value higher than 30 cycles based on the diagnostic PCR used at GOSH. Since $C_t$ values are only a surrogate for viral load and are not standardized across different assays (*Evans et al., 2021*), appropriate thresholds would need to be determined for other primary PCR testing assays. Similarly, variant calls at very low frequency were less likely to be present in both technical replicates. These variants at low frequency are thus potentially not genuine and the result of sequencing and variant calling errors. For our work, we removed any variants with an allele frequency lower than 2%. Until sequencing and variant calling technologies improve for low-frequency variants, technical replicates will remain essential for the study of pathogen within-host diversity in order to distinguish genuine variation from sequencing noise. The effect of this noise on phylogenetic inference will depend on

the signal-to-noise ratio and the amount of variation already present in the consensus sequences. Spurious low-frequency variation will likely affect only the branch length estimation in phylogenetic inference by adding potentially erroneous calls, unless there is presence of batch bias which could artificially cluster epidemiologically unrelated isolates together.

The generation, maintenance, and evolution of subpopulations within the host reflect evolutionary processes which are meaningful from phylogenetic and epidemiological perspectives. Subpopulations within a host can emerge from three mechanisms: de novo diversification in the host, transmission of a diverse inoculum, or multiple transmission events from different sources. If the subpopulations are the result of de novo mutations, nucleotide polymorphisms within the subpopulations accumulate over time and may therefore result in a phylogenetic signal useful for phylogenetic inference. In our data, longitudinal samples taken at later time points were demonstrated to accrue genomic variation. Although this pattern can be confounded by decreasing viral load as infection progresses, $C_t$ values in our dataset were not correlated with a higher genetic distance, and clusters in our data containing both longitudinal and technical replicates also corroborate these results. Transmission of a diverse inoculum also gives rise to phylogenetically informative shared low-frequency variants, as our results show that transmission pairs are more likely to share variants at low frequency. The effect of multiple transmission events in the phylogeny depends on the relatedness of both index cases and the bottleneck size in each transmission event.

Paired samples with epidemiological links and from the same department shared a higher proportion of low-frequency variants and were located closer in the consensus tree than samples with no relationship. These patterns suggest that the distribution of low-frequency variants is linked to events of epidemiological interest. The fact that technical duplicates shared more within-host diversity than longitudinal replicates of the same sample suggests that much of the variation within hosts is transitory. Therefore, within-host diversity may be relevant on relatively short time scales, which is precisely where consensus sequences lack resolution. Combining the data derived from fixed alleles in the consensus sequences and transient within-sample minor variation enables an improved understanding of the relatedness of pathogen populations between hosts.

The effects of neglecting within-host diversity in phylogenetic inference were analyzed by using simulated sequences under a phylogenetic model that reflects the presence and evolution of within-host diversity. We compared a conventional consensus phylogenetic model and a model that leverages within-sample diversity, and evaluated their ability to infer the known phylogeny. Our proposed phylogenetic model incorporates within-sample variation by explicitly coding major and minor nucleotides as independent characters in the alignment. We demonstrated that phylogenies inferred using the conventional consensus sequence approach were unresolved and unrepresentative of the known structure of the simulated tree. However, sequences that included within-host diversity showed higher resolution that resulted in phylogenetic trees more similar to the simulated phylogeny. As other mutational models, our 16-state model assumes independence between sites in the alignment. This assumption can be violated due to the presence of genetic linkage, which can be caused by multiple biological processes, such as clonal relationships between microorganisms, recombination, or selection of co-evolving sites. To increase the amount of genetic linkage due to clonal relationships between organisms, we repeated our simulations using a coalescent model to create the starting tree, and confirmed that the 16-state model still outperformed the conventional consensus sequence in the presence of high linkage. Other sources of genetic linkage are not accounted for, and their inclusion in phylogenetic inference is out of scope of this work.

The proposed phylogenetic model used for the simulations did not include direct base transitions and transversions, but rather modeled base change as a process of minor variant acquisition and fixation. Therefore, a base change is composed of the following steps: first a minor variant is gained; then the minor variant increases in frequency and becomes the majority variant; and finally the new variant is fixed. In this way, within-host evolution is partially included in the model as a process of minor variant evolution and eventual lost or fixation. As shown in the simulations, this process of within-host evolution is also captured when the minor bases are simply incorporated as additional states in the Markov chain without explicitly limiting the possible transitions.

We complemented the phylogenetic simulations with tree inference of outbreaks simulated using TransPhylo (*Didelot et al., 2017*). We parameterized the simulations to reflect the transmission dynamics of SARS-CoV-2, including a generation time of 5 days and a sampling time of 7 days.

Given this parameters, most simulated outbreaks lasted less than a month. We then simulated genetic sequences within the outbreak using fastsimcoal2 (*Excoffier et al., 2013*) as previously described by *De Maio et al., 2018*. Using a mutation rate of $5 \times 10^{-6}$ mutations per base per replication cycle, as previously described for SARS-CoV-2 and other betacoronaviruses (*Sender et al., 2021*; *Amicone et al., 2022*), and varying bottleneck sizes, we showed that tree inference using within-sample diversity improves as the transmission bottleneck widens, although even at low bottleneck sizes trees inferred using within-sample diversity are more accurate than those inferred using consensus sequences. Similarly, using varying mutation rates and a constant bottleneck size of 10 pathogens, we showed that tree inference was more accurate as mutation rates increased, although inference using consensus sequences improved only at a very high mutation rate of $10^{-3}$ mutations per base per cycle, which has mostly been observed in some HIV studies (*Cuevas et al., 2015*). Together, our simulations show that at the short time frame of disease transmission, phylogenetic inference using alignments that contain information regarding within-sample diversity outperform phylogenies inferred with consensus sequences, even at narrow transmission bottlenecks and very low mutation rates. Since TransPhylo simulates phylogenetic trees alongside the outbreak simulation, we could directly compare our inferred phylogenies with the known simulated trees. However, although phylogenetic trees can inform transmission inference, phylogenetic trees themselves and transmission trees are not interchangeable. Nevertheless, increasing the resolution of phylogenetic trees can improve inference of transmission chains and calculation of the likelihood of transmission events.

Previous studies have addressed the use of within-host variation to infer transmission events. *Wymant et al., 2018*, employed a framework based on phylogenetic inference and ancestral state reconstruction of each set of populations detected within read alignments using genomic windows. Our study extends this work by coding genome-wide diversity within the host directly in the alignment and the phylogenetic model. *De Maio et al., 2018*, proposed direct inference of transmission from sequencing data alongside host exposure time and sampling date within the Bayesian framework BEAST2 (*Bouckaert et al., 2014*). Our approach is focused on directly improving the temporal and phylogenetic signal of whole-genome sequences, and it's especially suited for use in applications and analysis that employ a phylogenetic tree as input to infer transmission (*Didelot et al., 2017*).

Apart from transmission inference, phylogenetic trees can be used to infer many parameters of epidemiological interest, such as $R_0$ or the effective population size. In our work, we showed that the temporal signal of clusters where all isolates had the same sequences increased with the inclusion of within-sample diversity, which in turns allows better inference of phylogenetic trees. When analyzing specific outbreaks, we showed that groups of samples without genetic differences were clustered apart from other isolates of the outbreak, providing additional information on genetic relationships that could be used for transmission inference or to better understand the genetic structure of the outbreak. Even though transmission inference can be improved with epidemiological data such as collection dates even when all isolates have the same genetic sequences, such data can't provide information regarding how samples cluster within the outbreak. Additionally, the order of collection dates not always correspond to the order of infection.

Future work will extend this model by including allele frequency data in addition to independent characters for major and minor variants. Moreover, to limit the number of character states we only allowed two variants at each position. Transmission inference of pathogens with high levels of within-host diversity, for instance as observed in HIV, could benefit from including more than two alleles. In those cases, the number of possible character state combinations would be too large, and therefore other methods such as phyloscanner (*Wymant et al., 2018*) could resolve transmission events more accurately. However, it's important to note that the low frequency of a third allele could result in more sequencing and mapping errors which could in turn bias phylogenetic inference and genomic analysis. Phylogenetic models that explicitly include dynamics of within-sample variation and sequencing error may further improve phylogenetic inference or allow researchers to better estimate parameters of interest, including R0, bottleneck size, transmissibility, and the origin of outbreaks.

In line with conventional consensus sequencing approaches, we used a reference sequence for genome alignment and variant calling. Although widely used, one limitation of this approach is a potential mapping bias causing some reads to reflect the reference base at low frequencies at a position where only a variant should be present. Although we applied stringent quality filtering, we cannot rule out the persistence of some false positive minor variants. Using genome graphs to map to

a reference that encompasses a wider spectrum of variation may alleviate this problem, and could be an interesting addition to pathogen population genomic analysis.

Our results demonstrate that within-sample variation can be leveraged to increase the resolution of phylogenetic trees and improve our understanding of who infected whom. Using SARS-CoV-2 hospital outbreaks and simulations, we show that variants at low frequencies are consistent within sample replicates, phylogenetically informative and are more often shared among epidemiologically related contacts. By coding within-sample variation directly in the alignment, the additional genetic information can be easily incorporated in phylogenetic inference, facilitating its application within existing epidemiology pipelines and public health infrastructure. We propose that pathogen phylogenetic models should accommodate within-host variation to improve the understanding of infectious disease transmission and aid infection control measures.

## Materials and methods
### Model for within-host diversity
To test the accuracy of different models at inferring known phylogenies, 100 random phylogenetic trees with 100 tips each were generated using the function *rtree* within the R package *ape* (*Paradis et al., 2004*). Whole-genome alignments were simulated from the random 100 phylogenies with the function *SimSeq* of the R package *phangorn* (*R Development Core Team, 2021*; *Schliep, 2011*) using a model with 16-character states that represent the combinations of the four nucleotides with each other as minor and major alleles (*Figure 3d*). Three substitution rates for the model were considered: a rate at which minor variants evolve, equal to 1; the rate at which minor variants are lost, leaving only the major nucleotide at that position, equal to 100; and the rate at which minor/major variants are switched, equal to 200. This rates result in fixed bases (A, C, G, and T) being 100 times more frequent than low-frequency bases. A different set of simulations was performed using rates that promote a high rate of low-frequency variation by having a lower rate of variant loss and switch (rates 1, 10, 10 for minor variant evolution, loss, and switch, respectively); a low amount of low-frequency variation by increasing the rates of variant switch and loss (1, 10, 100); and using a Jukes-Cantor model of sequence evolution and therefore resulting in no minor variants.

Two types of alignments were generated from the simulated genomes: a consensus sequence, where only the major allele was considered; and an alignment that retained the major and minor allele information as independent character states. RaxML-NG (*Kozlov et al., 2019*) was used to infer phylogenetic trees. The consensus sequence was analyzed with a GTR+$\gamma$ model, while the PROTGTR+$\gamma$ model was used for the alignment with intra-host diversity and major/minor variant information.

Several metrics were used to compare the 200 inferred phylogenetic trees with their respective starting phylogeny from which the sequences were simulated (*Supplementary file 3*). We chose metrics available in R suitable for unrooted trees, using the option "rooted = FALSE" where appropriate. The Robinson-Foulds distance (*Robinson and Foulds, 1981*) calculates the number of splits differing between both phylogenetic trees. For the weighted Robinson-Foulds, the distance is expressed in terms of the branch lengths of the differing splits. The Kuhner-Felsenstein distance (*Kuhner and Felsenstein, 1994*) considers the edge length differences in all splits, regardless of whether the topology is shared or not. Last, the Penny-Steel distance or path difference metric (*Steel and Penny, 1993*) calculates the pairwise differences in the path of each pair of tips, with the weighted Penny-Steel distance using branch length to compute the path differences. All functions were used as implemented in the package *phangorn* (*Schliep, 2011*) within R (*R Development Core Team, 2021*).

### Outbreak simulations
Disease outbreaks of size between 10 and 15 hosts were simulated using TransPhylo (*Didelot et al., 2017*), with a mean generation time of 5 days and a mean sampling time of 7 days, both parameters with standard deviation of 1 day (*Wang et al., 2020*; *Hart et al., 2022*). To ensure that the outbreak ends, the negative binomial distribution for the offspring number was set with a mean of 1 and a dispersion parameter of 0.5, resulting in a basic reproductive number (R$_0$) of 1. To simplify the simulations, all hosts from the outbreak were sampled. A total of 20 outbreaks were simulated. The population evolution within and between hosts was simulated using fastsimcoal2 (*Excoffier et al., 2013*) as previously described by *De Maio et al., 2018*, where transmissions are incorporated as population

migrations with a given bottleneck size and populations evolve with a given mutation rate per generation time. Sequences were simulated for a within-host population size of 1000 and a genome size of 1000 bp. To understand the effect of transmission bottleneck size in phylogenetic inference, varying values of bottleneck size were used along a constant mutation rate of $5 \times 10^{-6}$ mutations per base per generation cycle. Additionally, sequences were simulated at different mutation rates with a constant bottleneck size of 10 pathogens. Sequences with the varying bottleneck sizes and mutation rates were simulated using the same 20 simulated outbreaks. Phylogenetic trees were inferred from the alignments using RaxML-NG as previously described. The resulting trees were time-calibrated using the additive uncorrelated relaxed clock model (ARC) as implemented in BactDating (*Didelot et al., 2018*). The root of the outbreak was inferred as part of the dating model. The inferred trees were compared to the known simulated phylogenies using the Kuhner-Felsenstein distance (*Kuhner and Felsenstein, 1994*).

## Amplification and whole-genome sequencing

SARS-CoV-2 real-time qPCR confirmed isolates from London hospitals were collected as part of the routine diagnostic service at GOSH (*Storey et al., 2021*) and the COVID-19 Genomics UK Consortium (COG-UK) (*COVID-19 Genomics UK COG-UK, 2020*) between March and December 2020, in addition to epidemiological and patient metadata (*Supplementary file 2*). Multiple types of samples were collected: isolates from different patients; longitudinal replicates, where multiple isolates were collected from the same patient at different time points; and technical replicates, where multiple sequencing runs were performed from the same biological isolate. SARS-CoV-2 whole-genome sequencing was performed by UCL Genomics. cDNA and multiplex PCRs were prepared following the ARTIC nCoV-2019 sequencing protocol (*Tyson et al., 2020*). The ARTIC V3 primer scheme (*ARTIC Network, 2021*) was used for the multiplex PCR, with a 65°C, 5 min annealing/extension temperature. Pools 1 and 2 multiplex PCRs were run for 35 cycles. Five μL of each PCR were combined and 20 μL nuclease-free water added. Libraries were prepared on the Agilent Bravo NGS workstation option B using Illumina DNA prep (Cat. 20018705) with unique dual indexes (Cat. 20027213/14/15/16). Equal volumes of the final libraries were pooled, bead purified, and sequenced on the Illumina NextSeq 500 platform using a Mid Output 150 cycle flowcell (Cat. 20024904) (2×75 bp paired ends) at a final loading concentration of 1.1 pM.

## Whole-genome sequence analysis of SARS-CoV-2 sequences

Raw illumina reads were quality trimmed using Trimmomatic (*Bolger et al., 2014*) with a minimum mean quality per base of 20 in a 4-base wide sliding window. The 5 leading and trailing bases of each read were removed, and reads with an average quality lower than 20 were discarded. The resulting reads were aligned against the Wuhan-Hu-1 reference genome (GenBank NC_45512.2, GISAID EPI_ISL_402125) using BWA-mem v0.7.17 with default parameters (*Li and Durbin, 2010*). The alignments were subsequently sorted by position using SAMtools v1.14 (*Li et al., 2009*). Primer sequences were masked using ivar (*Grubaugh et al., 2019*).

Single-nucleotide variants were identified using the pileup functionality of samtools (*Li et al., 2009*) via the pysam package in Python (*Heger et al., 2023*). Variants were further filtered using bcftools (*Danecek et al., 2011*). Only variants with a minimum depth of 50× and a minimum base quality and mapping quality of 30 were kept. Additionally, variants within low complexity regions identified by sdust (*Li, 2019*) were removed. Previously identified problematic sites were masked to avoid systematic sequencing errors and phylogenetic bias (*De Maio et al., 2020*). For positions where only one base was present, the minimum depth was 20 reads, with at least 5 reads in each direction. Positions with low-frequency variants were filtered if the total coverage at that position was less than 100×, with at least 20 reads in total and 5 reads in each strand supporting each of the main two alleles.

Two different alignments were prepared from the data. First, an alignment of the consensus sequence where the most prevalent base at each position was kept. Variants where the most prevalent allele was not supported by more than 60% of the reads were considered ambiguous. Additionally, an alignment reflecting within-sample variation at each position as well as which base is the most prevalent and which one appears at a lower frequency by using the IUPAC nomenclature for amino acids (*IUPAC-IUB Joint Commission on Biochemical Nomenclature (JCBN), 1984*).

For the two different alignments, maximum likelihood phylogenies were inferred by using RAxML-NG (*Kozlov et al., 2019*) with 20 starting trees (10 random and 10 parsimony), 100 bootstrap replicates, and a minimum branch length of $10^{-9}$. For the consensus sequence, the GTR model was used. For the alignment reflecting within-host diversity, a model with amino acid nomenclature (PROTGTR) was used. All models allowed for a $\gamma$ distributed rate of variation among sites. Phylogenetic trees were time-calibrated using the known collection dates and the ARC model within Bact-Dating (*Didelot et al., 2018*). For transmission inference, the dated phylogeny was used with the longitudinal replicates removed by keeping the earliest sampled isolate. The likelihood of transmission was calculated using a SEIR model (*Lekone and Finkenstädt, 2006*; *Eldholm et al., 2016*).

## Code availability
All custom code used in this article can be accessed at https://github.com/arturotorreso/scov2_within-Host (copy archived at *Torres Ortiz, 2023*).

## Acknowledgements

The authors dedicate this article to the hospital staff members and patients who died of coronavirus disease 2019. They also thank all staff and patients who have taken part in the study. In addition, the authors are very grateful to the Great Ormond Street laboratory staff, the staff at the Camelia Botnar Laboratory, the Great Ormond Street Institute of Child Health and the COVID-19 sequencing team at UCLG who worked tirelessly to ensure that all PCR tests and sequencing work were completed in a timely manner during the COVID-19 pandemic. All authors acknowledge UCL Computer Science Technical Support Group (TSG) and the UCL Department of Computer Science High Performance Computing Cluster. LG was supported by the Wellcome Trust (201470/Z/16/Z), the National Institute of Allergy and Infectious Diseases of the National Institutes of Health under award number 1R01AI146338, and by the GOSH/ICH Biomedical Research Centre. XD was supported by the NIHR Health Protection Research Unit in Genomics and Enabling Data.

## Additional information

### Funding

| Funder | Grant reference number | Author |
|---|---|---|
| Wellcome Trust | 201470/Z/16/Z | Louis Grandjean |
| National Institute of Allergy and Infectious Diseases | 1R01AI146338 | Louis Grandjean |
| National Institute for Health Research | | Xavier Didelot |

The funders had no role in study design, data collection and interpretation, or the decision to submit the work for publication. For the purpose of Open Access, the authors have applied a CC BY public copyright license to any Author Accepted Manuscript version arising from this submission.

### Author contributions
Arturo Torres Ortiz, Conceptualization, Software, Formal analysis, Investigation, Visualization, Methodology, Writing – original draft, Writing – review and editing; Michelle Kendall, Formal analysis, Visualization, Methodology, Writing – review and editing; Nathaniel Storey, Investigation, Methodology, Writing – review and editing; James Hatcher, Helen Dunn, Sunando Roy, Rachel Williams, Data curation, Investigation, Writing – review and editing; Charlotte Williams, Data curation, Investigation, Methodology, Writing – review and editing; Richard A Goldstein, Supervision; Xavier Didelot, Formal analysis, Supervision, Investigation, Visualization, Methodology, Writing – review and editing; Kathryn Harris, Supervision, Investigation, Methodology, Writing – review and editing; Judith Breuer, Data curation, Writing – review and editing; Louis Grandjean, Resources, Data curation, Supervision, Investigation, Project administration, Writing – review and editing

## Author ORCIDs

Arturo Torres Ortiz http://orcid.org/0000-0002-2492-0958
Michelle Kendall http://orcid.org/0000-0001-7344-7071
Richard A Goldstein http://orcid.org/0000-0001-5148-4672
Xavier Didelot http://orcid.org/0000-0003-1885-500X
Judith Breuer http://orcid.org/0000-0001-8246-0534

## Decision letter and Author response

Decision letter https://doi.org/10.7554/eLife.84384.sa1
Author response https://doi.org/10.7554/eLife.84384.sa2

# Additional files

## Supplementary files

- Supplementary file 1. Study participants metadata.
- Supplementary file 2. Sample collection and demographics.
- Supplementary file 3. Metrics used for phylogenetic tree comparison.
- Supplementary file 4. Transition/transversion rates and base frequencies of the known simulated tree.
- Supplementary file 5. Inferred transition/transversion rates and base frequencies when using the consensus sequence. Numbers show the average of 100 simulations.
- Supplementary file 6. Inferred transition/transversion rates and base frequencies when accounting for within-host diversity. Numbers show the average of 100 simulations.
- MDAR checklist

## Data availability

Samples sequenced as part of this study have been submitted to the European Nucleotide Archive under accession PRJEB53224.Sample metadata is included in Supplementary file 1. All custom code used in this article can be accessed at https://github.com/arturotorreso/scov2_withinHost.git (copy archived at *Torres Ortiz, 2023*).

The following dataset was generated:

| Author(s) | Year | Dataset title | Dataset URL | Database and Identifier |
|---|---|---|---|---|
| Torres Ortiz A | 2022 | Within-host diversity improves phylogenetic reconstruction of SARS-CoV-2 outbreaks | https://www.ebi.ac.uk/ena/browser/view/PRJEB53224 | European Nucleotide Archive, PRJEB53224 |

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
