## [Editor Report]

This valuable study presents a novel and theoretically interesting model to account for viral diversity within hosts in evolutionary and genomic analyses of pathogens. The simulation results are solid and suggest that the model may provide new insight into SARS-CoV-2's and other pathogens' evolutionary dynamics.

---

## [Decision Letter]

**Decision letter after peer review:**

Thank you for submitting your article "Within-host diversity improves phylogenetic and transmission reconstruction of SARS-CoV-2 outbreaks" for consideration by *eLife*. Your article has been reviewed by 2 peer reviewers, and the evaluation has been overseen by a Reviewing Editor and Sara Sawyer as the Senior Editor. The following individuals involved in review of your submission have agreed to reveal their identity: Sebastian Duchene (Reviewer #1).

The reviewers have discussed their reviews with one another and with the Reviewing Editor, and the Reviewing Editor has drafted this to help you prepare a revised submission.

Essential revisions:

Both reviewers, while interested in the methodology, have raised concerns on the extent of the simulation study and on the validity of the conclusions based on analysis of the SARS-CoV-2 data.

Accordingly, the following are required for the paper to be considered further:

1) An expanded simulation study to explore in greater detail the assumptions and suitability of the proposed (16 site) model. Particular focal points, as raised by the reviewers (particularly #2, but discussed and agreed upon by both reviewers and the reviewing editor), would be on assumptions of Markovian behaviour and of site independence.

2) Address the concerns raised on the SARS-CoV-2 analyses. This may be achieved in a number of ways: a) [presumably supported by the additional work done through simulations] provide a stronger justification for the validity of the analyses as they are; or b) remove the SARS-CoV-2 analyses and either i) reframe the manuscript as a purely theoretical investigation; or ii) apply the model to a dataset with additional sequences (e.g. an influenza A virus dataset).

*Reviewer #2 (Recommendations for the authors):*

Lines 103-104: 2177x vs 10x: I don't fully understand.

Line 112: What does 'stable' mean? This is used repeatedly throughout the manuscript and I don't have a clear idea of what is meant by it.

Many of the y-axis labels and figure legends allude to 'mixed variants'. I searched for a definition of 'mixed' in the pdf but did not find one. I don't understand what is meant by 'mixed'.

Line 130: 'as viral load…also decreases from time since infection': this is not true.

Lines 147-152, Table 1: I'm somewhat confused. Line 141 talks about calculating the proportion of shared within-sample variants. Is that proportion used in this paragraph and for the values in Table 1? If so, how is 'phylogenetic distance' calculated from these proportions? I assume that what is used instead are the consensus sequences from each sample? But how are sampling dates incorporated? (They would seem to need to be incorporated if phylogenetic distance is indeed measured by substitutions per genome per year, and the phylogeny is time-aligned.) Also, I don't clearly understand the point of Table 1. If the point is simply to show that samples in one sample relationship (e.g., epidemiological) are genetically more similar that samples in a broader sample relationship (e.g., department), then why not show mean pairwise nucleotide differences between samples in a given sample relationship?

Table 1: Phylogenetic distance (or nucleotide distance) can't be negative, so the 95% CIs can't go below 0. That some of those distances have 95% CIs that go below 0 indicates to me that likely a normal distribution was used to calculate the 95% Cis when the distribution is not normal and some other distribution needs to be used (one that has no density for <0 values).

Figure 2: I assume the (%) on the y-axis label is an error and should be removed?

Figure 5: right-hand side panel: "subs/base/cycle" – do you mean mutations/base/cycle?

Line 324: "Our study benefited from…" This is a pretty substantial overstatement.

Line 376: can the parameters of the negative binomial be given as mean and overdispersion parameter (k)?

[Editors' note: further revisions were suggested prior to acceptance, as described below.]

Thank you for resubmitting your work entitled "Within-host diversity improves phylogenetic and transmission reconstruction of SARS-CoV-2 outbreaks" for further consideration by *eLife*. Your revised article has been evaluated by Sara Sawyer (Senior Editor) and a Reviewing Editor.

The manuscript has been significantly improved but there are some remaining issues that need to be addressed, as outlined below:

Following Reviewer #2's concerns (please see below), we ask that you more explicitly acknowledge the limitations of the SARS-CoV-2 data used in the study. This does not require further adjustments to the methods or analyses, but does require an earlier and stronger mention of the limitations, and (importantly) a discussion of *how* results may be impacted, not just that some of the limitations *do (likely)* impact them.

With regards to Reviewer #2's query on Figure 2 (also see their original review), it does remain of concern and must be addressed. While the additional material shown in Supp Figure 5 is helpful (as it is clear there that there are substantial and statistically meaningful differences there), a claim that "the proportion of shared within-host variants was significantly higher…" (line 171) requires further (statistical and/or compelling visual) justification. The current text at lines 168--187 is rather unclear as it is hard to tell if "shared variants" or "shared low frequency variants" are being described. If the 95% CIs shown in Figure 2 have been correctly calculated, then statistical (and/or compelling visual) analyses that justify the claim need to be clearly reported.

*Reviewer #2 (Recommendations for the authors):*

I have reviewed the revised manuscript, which has been improved in terms of testing the 16-site model on simulated data and on more clearly defining terms. My remaining concerns are:

1. The manuscript interprets its data and its findings too favorably. It would be much improved if the authors carefully went through the manuscript and improved the accuracy of statements/conclusions throughout. For example, the subtitle at line 151 reads 'Within-sample variation is shared between technical replicates'. Yet Figure 1c indicates that this is an overstatement. The proportion of shared minor variants is actually highly variable between technical replicates. A more appropriate subtitle would be 'Some within-sample…' Similarly, line 147, a more accurate subtitle would be 'A small amount of within-sample variation appears to be shared between…'

2. Even by selecting only the subset of samples with Ct < 30, the overwhelming majority of the SARS-CoV-2 data minor variants in this subset are still spurious, given the cross-replicate findings shown in Figure 1. (The samples with lower Ct have a higher proportion of minor variants shared but they also have fewer variants, so that when the data subset of 249 samples are analyzed, the overwhelming majority of the minor variants in that dataset will be spurious or at least not reproducible across technical replicates.) This needs to be acknowledged directly in the manuscript and implications of this for their findings need to be discussed. The discussion currently has starting on line 408 an acknowledgment that many of the minor variants in their SARS-CoV-2 samples are likely not genuine. However, there is not a discussion of how that could have impacted their results (shown in Figure 6 and 7), and that acknowledgment does not come out clearly in the Results section of the text.

3. Paragraph 155-161: why not just look at Hamming distance? (Why go through a tree?) This result is also expected and I am not sure how it plays into the main point of this paragraph that within-sample variation is shared between epidemiologically linked samples.

4. Lines 174-176: I am trying to still square this with the data shown in Figure 2 and the confidence intervals shown in this figure. To be more convincing that the probability of a variant being shared is actually higher in epidemiologically-linked samples than is unlinked samples, could the authors show the specific instances of when they are shared? Again, I think showing a plot with one sample's minor variant frequencies shown on the x-axis and another sample's minor variant frequencies are shown on the y-axis would be highly informative. The plot does not have to identify (or assume) donor or recipient, it just would need to show the frequencies of the shared minor variant in both samples, along with the frequencies of all other minor variants that were identified in those samples (and any other sites where the alleles might be fixed but differ). Plots like these would be helpful to visualize patterns of sharing of minor variations.

5. Simulation study starting at line 191: I like this addition but am confused about what this simulation study entails. How did you simulate the genome alignments? If it is from the 'Model for within-host diversity' described in Materials and methods (which I suggest renaming, since the model simulates within- and between-host evolution if I am understanding correctly), then this model also has linkage between nucleotide sites. (Lines 228-233 present a coalescent model, which they say accounts for linkage, but this first model also has full linkage as far as I can tell.) Also, I believe the previous issue was less about linkage per se, it was about independence of transitions across sites. For example, during a transmission event with a bottleneck size of 1, all sites with genetic variation (e.g. Cg, At, Ta…) would become monomorphic at the same time (e.g., C, A, T), and the 16-site model assumes independence of these transitions across sites.

6. Figure 1—figure supplement 2: the smoothing has made there be some density of samples that have negative mean coverage, which doesn't quite make sense.

---

## [Author Response]

Essential revisions:Reviewer #2 (Recommendations for the authors):Lines 103-104: 2177x vs 10x: I don't fully understand.

This sentence has been clarified (now Line 110): “A total of 454 whole-genomes with mean coverage higher than 10x were kept for further analysis, resulting in an average coverage across isolates of 2457x (Figure 1—figure supplement 2).”

Line 112: What does 'stable' mean? This is used repeatedly throughout the manuscript and I don't have a clear idea of what is meant by it.

We agree with the reviewer that ‘stable’ is a somewhat loose term. We have changed all instances of ‘stable’ for ‘shared’ or ‘consistent’ throughout the manuscript.

Many of the y-axis labels and figure legends allude to 'mixed variants'. I searched for a definition of 'mixed' in the pdf but did not find one. I don't understand what is meant by 'mixed'.

We have changed all instances of “mixed variant” for “minor variant” for clarity. We also defined “minor variant” in the Results section (Line 113).

Line 130: 'as viral load…also decreases from time since infection': this is not true.

We have rephrased this sentence to be more accurate (now Line 137): “or due to the accumulation of within-host variation through time, as late stages of infection are usually characterized by high Ct values (low viral load).”

Lines 147-152, Table 1: I'm somewhat confused. Line 141 talks about calculating the proportion of shared within-sample variants. Is that proportion used in this paragraph and for the values in Table 1? If so, how is 'phylogenetic distance' calculated from these proportions? I assume that what is used instead are the consensus sequences from each sample? But how are sampling dates incorporated? (They would seem to need to be incorporated if phylogenetic distance is indeed measured by substitutions per genome per year, and the phylogeny is time-aligned.) Also, I don't clearly understand the point of Table 1. If the point is simply to show that samples in one sample relationship (e.g., epidemiological) are genetically more similar that samples in a broader sample relationship (e.g., department), then why not show mean pairwise nucleotide differences between samples in a given sample relationship?

We agree with the reviewer that the section referenced was structured in a confusing way. Moreover, the legend of Table 1 specified that the genetic distance was measured in substitutions per genome per year, which is not correct. We have restructured the paragraph in a more clear way and we have changed the legend to reflect that the genetic distance is measured in units of substitutions per genome (also reflected in the text now), and that the distance is calculated using the ML consensus phylogenetic tree.

For our purposes, both the phylogenetic distance and the raw SNP distance are useful measures, and in principle they should be similar, but as pointed out by the reviewer the raw genetic distance is a more simple metric. We have changed Table 1 where we have used the pairwise SNP distance between isolates.

Table 1: Phylogenetic distance (or nucleotide distance) can't be negative, so the 95% CIs can't go below 0. That some of those distances have 95% CIs that go below 0 indicates to me that likely a normal distribution was used to calculate the 95% Cis when the distribution is not normal and some other distribution needs to be used (one that has no density for <0 values).

We have changed the model to a more appropriate Γ distribution.

Figure 2: I assume the (%) on the y-axis label is an error and should be removed?

The reviewer is correct in pointing out that the proportion is not a percentage. We have removed the “%” symbol.

Figure 5: right-hand side panel: "subs/base/cycle" – do you mean mutations/base/cycle?

We have changed the x-axis to mutations/base/cycle as pointed out by the reviewer.

Line 324: "Our study benefited from…" This is a pretty substantial overstatement.

We have updated this paragraph in Lines 408 to 412 for a more suitable exploration of how technical replicates benefited the study.

Line 376: can the parameters of the negative binomial be given as mean and overdispersion parameter (k)?

We have changed Line 376, which now reads: “To ensure that the outbreak ends, the negative binomial distribution for the offspring number was set with a mean of 1 and a dispersion parameter of 0.5, resulting in a basic reproductive number (R0) of 1.”

[Editors' note: further revisions were suggested prior to acceptance, as described below.]

The manuscript has been significantly improved but there are some remaining issues that need to be addressed, as outlined below:Following Reviewer #2's concerns (please see below), we ask that you more explicitly acknowledge the limitations of the SARS-CoV-2 data used in the study. This does not require further adjustments to the methods or analyses, but does require an earlier and stronger mention of the limitations, and (importantly) a discussion of *how* results may be impacted, not just that some of the limitations *do (likely)* impact them.

We have now updated the beginning of the Discussion section, more specifically the second paragraph, to highlight the effects of spurious variant calls on phylogenetic inference.

With regards to Reviewer #2's query on Figure 2 (also see their original review), it does remain of concern and must be addressed. While the additional material shown in Supp Figure 5 is helpful (as it is clear there that there are substantial and statistically meaningful differences there), a claim that "the proportion of shared within-host variants was significantly higher…" (line 171) requires further (statistical and/or compelling visual) justification. The current text at lines 168--187 is rather unclear as it is hard to tell if "shared variants" or "shared low frequency variants" are being described. If the 95% CIs shown in Figure 2 have been correctly calculated, then statistical (and/or compelling visual) analyses that justify the claim need to be clearly reported.

Figure 2 shows a basic summary of the raw data, with the mean and 95th and 5th percentiles. We agree the text was confusing by using the term 95% CI rather than percentiles, and this has been amended. The modeling is shown in Figure 2—figure supplement 2, where the shaded area does represent 95%CI and where p-values were calculated. This has been clarified in the text. We have also added Figure 2—figure supplement 1 as requested by Reviewer #2 to show the differences in allele frequency between samples with different links. We have also changed the text in 168-187.

Reviewer #2 (Recommendations for the authors):I have reviewed the revised manuscript, which has been improved in terms of testing the 16-site model on simulated data and on more clearly defining terms. My remaining concerns are:1. The manuscript interprets its data and its findings too favorably. It would be much improved if the authors carefully went through the manuscript and improved the accuracy of statements/conclusions throughout. For example, the subtitle at line 151 reads 'Within-sample variation is shared between technical replicates'. Yet Figure 1c indicates that this is an overstatement. The proportion of shared minor variants is actually highly variable between technical replicates. A more appropriate subtitle would be 'Some within-sample…' Similarly, line 147, a more accurate subtitle would be 'A small amount of within-sample variation appears to be shared between…'

We thank the reviewer for the time and throughout the comments. We have now changed all the subheadings to neutral statements to avoid any interpretation of the data in the Results section.

2. Even by selecting only the subset of samples with Ct < 30, the overwhelming majority of the SARS-CoV-2 data minor variants in this subset are still spurious, given the cross-replicate findings shown in Figure 1. (The samples with lower Ct have a higher proportion of minor variants shared but they also have fewer variants, so that when the data subset of 249 samples are analyzed, the overwhelming majority of the minor variants in that dataset will be spurious or at least not reproducible across technical replicates.) This needs to be acknowledged directly in the manuscript and implications of this for their findings need to be discussed. The discussion currently has starting on line 408 an acknowledgment that many of the minor variants in their SARS-CoV-2 samples are likely not genuine. However, there is not a discussion of how that could have impacted their results (shown in Figure 6 and 7), and that acknowledgment does not come out clearly in the Results section of the text.

The reviewer’s point that removing samples with a high Ct will still keep a big proportion of spurious minor variants is correct. That is why we also filtered variants with an allele frequency lower than 2%. We realize this was not explained in the Results section, which we have now corrected in the paragraph starting in line 140. We apologize for any confusion. Figure 2 reflects the probability a variant is shared for the filtered dataset.

We have also included a paragraph at the beginning of the Discussion about the presence and effects of spurious minor variation in the data.

3. Paragraph 155-161: why not just look at Hamming distance? (Why go through a tree?) This result is also expected and I am not sure how it plays into the main point of this paragraph that within-sample variation is shared between epidemiologically linked samples.

We apologize as we did change the phylogenetic distance to a hamming (SNP based) distance in Table 1, but this was not properly addressed in the paragraph the reviewer was referring to. We have now changed the paragraph starting at line 155 to reflect the reviewers’ comment. We believe it is important to analyze the concordance between the genetic and the epidemiological data before analyzing how within-host variants are shared among epidemiologically linked samples.

4. Lines 174-176: I am trying to still square this with the data shown in Figure 2 and the confidence intervals shown in this figure. To be more convincing that the probability of a variant being shared is actually higher in epidemiologically-linked samples than is unlinked samples, could the authors show the specific instances of when they are shared? Again, I think showing a plot with one sample's minor variant frequencies shown on the x-axis and another sample's minor variant frequencies are shown on the y-axis would be highly informative. The plot does not have to identify (or assume) donor or recipient, it just would need to show the frequencies of the shared minor variant in both samples, along with the frequencies of all other minor variants that were identified in those samples (and any other sites where the alleles might be fixed but differ). Plots like these would be helpful to visualize patterns of sharing of minor variations.

We have added Figure 2—figure supplement 1, where the allele frequencies of isolates with different relationships are compared in a pairwise manner. We have also changed the legend in Figure 2 to reflect that the error bars actually represent 95th and 5th percentile, not confidence intervals. Confidence intervals are shown in the modeling of Figure 2—figure supplement 2.

5. Simulation study starting at line 191: I like this addition but am confused about what this simulation study entails. How did you simulate the genome alignments? If it is from the 'Model for within-host diversity' described in Materials and methods (which I suggest renaming, since the model simulates within- and between-host evolution if I am understanding correctly), then this model also has linkage between nucleotide sites. (Lines 228-233 present a coalescent model, which they say accounts for linkage, but this first model also has full linkage as far as I can tell.) Also, I believe the previous issue was less about linkage per se, it was about independence of transitions across sites. For example, during a transmission event with a bottleneck size of 1, all sites with genetic variation (e.g. Cg, At, Ta…) would become monomorphic at the same time (e.g., C, A, T), and the 16-site model assumes independence of these transitions across sites.

The simulations described in line 191 were carried out using the Within-host diversity model described in the Results and Methods, as pointed out by the reviewer. This is a hidden-markov model with 16 states, thus modeling some of the processes that create within-host diversity.

The reviewer is correct by pointing out that the Within-host diversity model already has linkage within it. We carried out additional simulations using a coalescent tree rather than a random tree as a starting point to increase linkage across sites compared to a random starting tree to understand how the 16-state model fared when high amounts of linkage were present.

We also agree with the reviewer that linkage and independence of transitions is a biological phenomenon that is not accounted for in our model or in any other mutational models. The scenario suggested by the reviewer, where whole genotypes are in linkage and therefore transitions occur all at once when one population in particular increases in frequency (either by selection or drift, as in the case of a transmission bottleneck), is a phenomenon that will occur independently of the chosen model, regardless of whether the model accounts for 4- or 16-states. This is a topic of paramount importance, and hard to model in microbial phylogenetics, but out of the scope of the topic of this paper. However, we have clarified this limitation in the Discussion section, starting in line 354.

6. Figure 1—figure supplement 2: the smoothing has made there be some density of samples that have negative mean coverage, which doesn't quite make sense.

We have changed Figure 1—figure supplement 2 for a histogram to reflect better the distribution of mean coverage.